# Novel Efficient Physical Technologies for Enhancing Freeze Drying of Fruits and Vegetables: A Review

**DOI:** 10.3390/foods12234321

**Published:** 2023-11-29

**Authors:** Jianhua Yao, Wenjuan Chen, Kai Fan

**Affiliations:** 1College of Life Science, Yangtze University, Jingzhou 434025, China; 2National Polymer Materials Industry Innovation Center Co., Ltd., Guangzhou 510530, China; 3Institute of Food Science and Technology, Yangtze University, Jingzhou 434025, China

**Keywords:** physical fields, quality, freeze drying, fruits and vegetables

## Abstract

Drying is the main technical means of fruit and vegetable processing and storage; freeze drying is one of the best dehydration processes for fruit and vegetables, and the quality of the final product obtained is the highest. The process is carried out under vacuum and at low temperatures, which inhibits enzymatic activity and the growth and multiplication of micro-organisms, and better preserves the nutrient content and flavor of the product. Despite its many advantages, freeze drying consumes approximately four to ten times more energy than hot-air drying, and is more costly, so freeze drying can be assisted by means of highly efficient physical fields. This paper reviews the definition, principles and steps of freeze drying, and introduces the application mechanisms of several efficient physical fields such as ultrasonic, microwave, infrared radiation and pulsed electric fields, as well as the application of efficient physical fields in the freeze drying of fruits and vegetables. The application of high efficiency physical fields with freeze drying can improve drying kinetics, increase drying rates and maintain maximum product quality, providing benefits in terms of energy, time and cost. Efficient physical field and freeze drying technologies can be well linked to sustainable deep processing of fruit and vegetables and have a wide range of development prospects.

## 1. Introduction

Drying is a dehydration process that is widely used, and has been one of the most popular methods of food preservation in the food processing and preservation industry for many years, where the volume of water is reduced and the main aim is to achieve long-term preservation of food by reducing the water activity in the cells [1]. The drying process of food prevents microbial contamination and chemical changes such as enzymatic and non-enzymatic browning, prolongs the storage life, and reduces packaging, handling and transport costs [2]. The drying process has been used in a wide range of food products such as meat, rice, pasta, dairy, fruit and vegetable products.

Common drying methods include hot-air drying [3], microwave drying [4], heat pump drying [5], infrared radiation drying [6] and freeze drying [7]. Each technique of drying can be chosen depending on the size, structure, color, composition, nutrients, maturity and desired effect of the products. Different drying methods exhibit various drying characteristics during drying. Freeze drying (FD) is the removal of ice from a sample by sublimation to achieve a dehydrating effect. Freeze-dried products are considered to have the same characteristics as fresh products, and freeze drying has become one of the most attractive and applicable methods for food ingredients [8]. In comparison to other drying techniques, freeze drying allows maximum retention of the color, appearance, texture and flavor of fresh samples and avoids loss of nutrients [3]. There is a high rehydration capacity and the interior is spongy and porous [9]. The mango powder obtained after freeze drying has a similar color to mango pulp and retains a better rehydration, allowing it to be used in many food formulations [10]. Under appropriate operating conditions, freeze drying is an alternative to dehydration of yellow passion fruit residues, where bioactive substances such as total phenols, flavonoids and pectin are increased after dehydration, reducing their loss [11]. Freeze-dried samples show minimal reduction in free radical scavenging, the microstructure remains regular and the protein powder maintains particle integrity [12]. However, the energy requirements are excessive, due to the three phase changes involved in the freeze drying process (mainly in the liquid, solid and gaseous states), and the long drying times and high running costs required. Freeze drying is considered to be the most expensive operation for producing dehydrated products, which limits its application.

The current market demand for high quality of dried products is an increasing trend, maintaining good organoleptic qualities and high nutritional content. Researchers are trying to find ways for improving the shortcomings of various drying methods, and the use of efficient physical techniques in freeze drying is gradually emerging. This paper reviews the applications of ultrasound assisted atmospheric pressure freeze drying, ultrasound pretreatment assisted freeze drying, infrared freeze drying, pulsed electric field assisted freeze drying and microwave freeze drying in fruits and vegetables, discusses their mechanisms of action and effects on drying kinetics and quality of fruits and vegetables based on available literature data, explores the problems of freeze drying and presents challenges for future development.

## 2. Freeze Drying

### 2.1. Principle of Freeze Drying

Water exists in three different states: liquid, solid and gaseous (steam). The basic principle of freeze drying lies in the three-state change of water. Figure 1 shows a graph of the relationship between temperature, pressure and the change of water phase; the curve is the process of the mutual transformation of solid, liquid and gaseous states. The temperature at the three-phase equilibrium point is 0.0098 °C and the pressure is 0.612 kPa, where the three phases of water (liquid, ice and steam) coexist [13]. Point C is the critical point for water (374 °C, 22,060 kPa), freezing of the water occurs when the temperature falls below 0.01 °C and water sublimates directly when the pressure falls below 0.612 kPa. Freeze drying is based on the principle that when the temperature is lowered, the water freezes and as the water vapor pressure decreases and heat is supplied sublimation is completed, resulting in a freeze-dried product.

Freeze drying is a drying technique in which food is frozen at low temperatures and the moisture in the material is directly removed by sublimation in a vacuum environment. It is generally divided into three stages: freezing, primary drying stage and secondary drying stage which, in turn, contains five activities: freezing, sublimation, desorption, vacuum extraction and steam condensation. Figure 2 illustrates several processes of freeze drying.

In the freezing stage, the food is cooled to the temperature of the frozen state, where the free water in the food is frozen into a solid [14]. This stage has a certain influence on the size and morphology of the ice crystals; the faster the drying rate, the smaller the ice crystals formed and the less influence on the structure of the sample. Conversely, larger ice crystals are produced and the sample forms a larger pore-like structure, which facilitates sublimation and increases the drying rate, but too large ice crystals can damage the cell structure, so freezing is a key part of the freeze drying process [15].

**Figure 2 foods-12-04321-f002:**
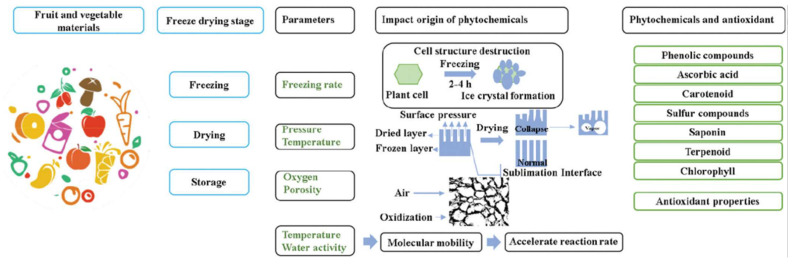
A typical freeze drying process, including freezing, primary drying and secondary drying (the data were adopted from production type vacuum freeze dryer) [16]. Reproduced with permission from copyright owner; published by Taylor & Francis, 2023.

In the sublimation drying stage, the solid ice formed by free water freezing in the sample is turned into water vapor by sublimation, removing approximately 90–95% of the water from the fruit and vegetables [9]. It is worth noting that the temperature profiles are dependent on sample geometry and the heat transfer mechanisms (contact/conduction/radiation). During the drying process, the temperature in different regions of the sample varies with the position of the sample and decreases from the surface to the sublimation interface [17]. The temperature of the sublimation surface has an important influence on the microstructure as well as the function of the sample, and if it is higher than the eutectic point, the pore walls become softer, leading to the closure of the channels and the problem of sample melting. Sublimation drying is actually a time-consuming and energy-intensive process, typically up to 8–9 h.

Desorption drying is completed after sublimation drying, the stage is relatively slow, fruits and vegetables in the internal porous structure of the residual bonded water in the high temperature conditions desorption to form free water, free water at high temperatures in the form of water vapor diffusion to the outside of the fruit and vegetables [13]. In addition to this, a higher temperature is required to complete the desorption process, but the temperature should not be too high and needs to be strictly controlled, requiring a temperature below the glass transition temperature of the foodstuff, otherwise fruits and vegetable will lead to denaturation and an eventual residual of 0.5–3% moisture [9]. The glass transition temperature (Tg) is the temperature at which a material changes from a glassy state to a rubbery state at a given heating rate, and Tg can be used as a reference parameter to characterize the nature, quality, stability and safety of food systems [18].

### 2.2. Mathematical Modelling of Freeze Drying

The simulation of the drying process of fruits and vegetables can be realized through the drying kinetic model, and the law of change in internal moisture and diffusion of fruits and vegetables in the drying process reflect the dehydration ability of the method. At present, researchers have carried out relevant mathematical model research on many food products, but in fruits and vegetables a series of physicochemical changes will occur in the drying process, as well as in the special organizational structure fruits and vegetables themselves. The previous mathematical model may have limitations, so the study of the drying kinetic model of optimization of the drying process is of great significance. In drying modelling, it is possible to achieve a link between the parameters of the drying process to the extent that the drying process can be accurately predicted and controlled, and reports on kinetic modelling of drying of fruits and vegetables are constantly on the rise. Among them, Sandall et al. [19] firstly proposed a mathematical model of freeze drying in 1967, in which he believed that the sublimation interface gradually receded uniformly to the freezing zone, and the model could only simulate the sublimation process of the samples with uniform tissue structures, which was known as the ice-crystal interface uniformly receded freezing model. In addition, quasi-steady-state models and sublimation resolution models have been proposed by researchers, and some of them have been improved on the basis of the above models. Nakagawa et al. [20] developed a simple mathematical model by assuming a quasi-steady-state energy balance at the sublimation interface, which consists of the classical heat–mass transfer equation, and performed ice sublimation experiments using apple blocks. Equations (1) and (2) are the heat and mass transfer equations in drying, respectively. Chaurasiya et al. [21] present a model relating to the description of heat–mass transfer for sublimation in semi-porous spaces that takes into account volumetric heat sources for convective heat–mass transfer in the drying and freezing zones, as well as convective currents driven by mass transfer from ice crystals. Equations (3) and (4) are mathematical formulas for the sublimation process, respectively. Srisuma et al. [22] present exact and approximate analytical solutions of the microwave-assisted freeze drying mechanistic model, which is also applicable in conventional and hybrid freeze drying; the exact solution obtained by the principle of superposition, the separation of variables and Duhamel’s theorem, and the approximate solution obtained by the thermal equilibrium integral.
$$T_{sh}=\frac{T_{Rsh}+\alpha T_{s}}{1+\alpha }$$
(1)$$\alpha =\frac{1/h_{Rsh}}{\frac{1}{A_{btm}h_{btm}}+\frac{\left(\frac{1}{\sqrt{\pi A_{btm}}}\right)-(\frac{1}{\sqrt{\pi A_{s}}})}{2\lambda_{dry}}}$$
(2)$$\frac{dm}{dt}=\frac{Mw}{R}\frac{\frac{Ps}{Ts}-Pc/Tc}{\frac{\left(\frac{1}{\sqrt{\pi As}}\right)-(\frac{1}{\sqrt{\pi Aext}})}{2D_{A}}}$$
where *T_sh_* is the temperature of shelf (basket) (K), *T_s_* is the temperature at sublimation interface (K), *h_sh_* is the heat transfer coefficient at the bottom of the product (W/m^2^ K), *A_btm_* is the surface area of frozen layer contacting the sample tray, *A_s_* surface area of sublimation interface (m^2^), *h_btm_* is the overall heat transfer coefficient at the frozen bottom of the product (W/m^2^ K), *k_dry_* is the thermal conductivity of dried layer (W/m K), *M_W_* is the molecular weight of water (kg/mol), *P_s_* is the pressure at sublimation interface (Pa), *P_c_* is the pressure at condenser surface (Pa), *A_ext_* is the external surface area of product contacting to ambience (m^2^), *D_A_* is the effective diffusion coefficient in dried layer (m^2^/s).

Frozen region (convective heat transfer): (3)$$\frac{1}{\alpha_{1}}\left(\frac{\partial T_{1}}{\partial t}+u_{1}\frac{\partial T_{1}}{\partial x}\right)=\frac{\partial^{2}T_{1}}{\partial x^{2}}+A_{1}(x,t),\;s\left(t\right)<x<\infty .$$

Dried region (convective heat/mass transfer): (4)$$\frac{1}{\alpha_{1}}\left(\frac{\partial T_{2}}{\partial t}+u_{2}\frac{\partial T_{2}}{\partial x}\right)=\frac{\partial^{2}T_{1}}{\partial x^{2}}+(\frac{C_{pw}}{k_{2}}\frac{dw}{dt})\frac{\partial T_{2}}{\partial x}A_{2}(x,t),\;0<x<s\left(t\right).$$
where *α* is thermal diffusivity (m^2^ s^−1^), *u* is the uni-directional molecular motion (m s^−1^), *t* is time (s), *A* is the volumetric heat source term (K m^−2^), *T* is temperature (K), *x* is space coordinate (m), *k* is the thermal conductivity (W m^−1^ K^−1^), *c_pw_* is the specific heat of water vapor (J kg^−1^ K^−1^), and 0, 1, and 2 in the subscripts represent initial, frozen, and dried, respectively.

## 3. Physical Fields

### 3.1. Ultrasound Technology

In the 1950s, it was first reported that sound waves could accelerate drying. At that time, research into ultrasound was interrupted because of the poor capacity of the equipment, and it was only at the beginning of the 21st century, when new ultrasound equipment became available, that researchers renewed their interest in this direction of research [23]. Ultrasound covers a much wider range of frequencies (above 100 kHz) and intensities (below 1 W/cm^2^). The use of ultrasound in academic research has become progressively more widespread and successful, but practical applications in food production are limited to niche products and need to be further expanded and developed. The use of ultrasound in the dehydration of fruits and vegetables enhances heat transfer, increases drying rates and reduces energy consumption. In the drying of fruit and vegetables, ultrasound is generally present in two ways: one as a pretreatment, and the other as a secondary processing method in conjunction with the drying method [24].

Figure 3a is a sketch of the ultrasound probe application. Ultrasonic treatment is based on mechanical, cavitation and thermal effects, and is a modern and evolving technology in the food industry [25,26,27].The ultrasound waves are transmitted through the ultrasound probe into the aqueous medium. At this point, the medium is compressed and expanded by the ultrasonic waves, just as a sponge is repeatedly pressed and released. On the one hand, the medium inside the sample is compressed and stretched, the forces between the moisture and the sample are weakened and percolation is formed inside the sample; on the other hand, changing the cellular structure of the sample creates microscopic channels in the sample where the intracellular fluid enters the aqueous medium and speeds up the diffusion of the intracellular fluid outwards [28]. The effectiveness of this action depends on the structure of the sample, with foods that are thin, soft and have porous tissue being more likely to benefit, such as eggplant [29,30]. The mechanism is mainly due to ultrasonic cavitation. When the ultrasonic power is too high, cavitation bubbles are generated, which gradually expand in size and rupture when they reach a critical value and release a large amount of energy, causing rapid changes in pressure and temperature, rupture of the cell membrane and destruction of the microstructure, leading to the decomposition of water molecules into hydroxyl radicals [31,32,33,34]. Cavitation effects can expand the internal pores of fruits and vegetables, improving the migration speed of water in the drying process, which can effectively strengthen the drying process and shorten the time of freeze drying [35]. Finally, ultrasound produces agitation as it propagates, and the individual media generate friction, which raises the temperature and provides heat for the food drying process, facilitating heat and mass transfer.

### 3.2. Infrared Radiation

Infrared radiation is located in the outer range of visible red light, and is an electromagnetic wave with a wavelength in the range 0.78–1000 μm. It can be divided into three categories: near infrared (NIR) (0.76–2 μm), mid infrared (MIR) (2–4 μm) and far infrared (FIR) (4–1000 μm) [37]. Each of the three zones is adapted to different samples, with NIR being more efficient for thicker samples and FIR penetrating 2–5 mm, producing better results for thin, fast drying and high moisture samples [38]. Figure 3b show a schematic diagram of IR drying.

Infrared radiation acts on the surface of the food, the internal moisture molecules match the vibration frequency of the infrared radiation and resonate with the infrared radiation energy to absorb it. The absorbed energy is converted into vibrations within the food molecules, and the radiant energy is converted into heat energy conducted into the food. The internal temperature of the food rises, and the moisture migrates from the inside to the outside, finally achieving the purpose of drying. The absorption of radiant energy is related to the wavelength of the infrared radiation and the characteristics of the food [39].

Infrared drying is now widely used in fruit and vegetable drying. Qiu et al. [40] has used near-infrared spectroscopy on roses in order to predict the antioxidant capacity of infrared dried roses and to improve drying efficiency. Alaei et al. [41] has simulated the process of drying pomegranates using a combination of vacuum and NIR drying. The Aghbashlo model is superior in predicting the drying behavior of pomegranate fruit. NIR drying is also used for olive leaves [42] and melons [43]. Onwude et al. [44] used a mid-infrared unit (2.5–4 μm) for drying sweet potatoes to study the drying kinetics, mass and heat transfer. The total drying time was reduced by 15% with an increase in IR intensity from 1100 W/m^2^ to 1400 W/m^2^. Wang et al. [45] used mid-infrared technology to dry daylilies. The results showed that compared to the far infrared, the mid-infrared treatment resulted in a 50% reduction in drying time, a 103% increase in the effective moisture diffusion rate and a 10% reduction in drying activation energy. Nathakaranakule et al. [46] used far-infrared combined with a hot-air device to dry longan, increasing the drying rate by shortening the drying time. Fernando et al. [47] investigated the drying characteristics and process parameter optimization of turmeric by far infrared drying, and found that far infrared radiation is effective for drying turmeric and determines that the optimum conditions for drying turmeric are a wavelength of 5.15 μm and a drying time of 21 min. In addition, far-infrared drying is used in buttercup [48], apple slices [16], rhizoma dioscoreae [49], cistanche [50] and potatoes [51].

In recent years, researchers have combined freeze drying and infrared [52,53,54]. The advantages of infrared radiation and FD are reflected in the application of infrared freeze drying, which is superior to single FD both in terms of drying efficiency and quality characteristics of fruits and vegetables, and the combined drying will result in a product with better rehydration and a darker color.

### 3.3. Pulsed Electric Field

Pulsed electric field (PEF) is an emerging non-thermal processing technique in which a voltage pulse is applied between two electrode plates so that it acts on the food between them for a short duration and generates no or very little heat. Around the middle of the twentieth century, PEF began to be applied in the processing of food and agricultural products. PEF technology is used extensively in the food processing industry for food sterilization, food preservation, extraction of compounds and food drying. PEF is also used in the production of potato and vegetable fries; in the process of French fries, PEF can reduce the consumption of water and energy, shorten the processing time, increase the internal porosity of fried potatoes and accelerate the evaporation of water vapor [55]. After pretreatment of potatoes, it will reduce the content of reducing sugar in the finished potato chips and reduce the occurrence of browning phenomenon [56]. In addition, PEF has a better cutting effect on potatoes, making the cut surface of potatoes smoother [55,57]. Figure 3c shows a schematic diagram of the application of a pulsed electric field.

The electroporation produced by PEF makes it a good pre-treatment for fruits and vegetables before drying. PEF accelerates water transport in cells mainly by enhancing the permeability of cell membranes, accelerating drying efficiency without changing the physicochemical properties of fruits and vegetables. The technical mechanism of PEF as a pre-treatment for freeze drying is not yet precisely explained, but studies have been carried out on this subject. It can be explained by the phenomenon of electroporation. When high-voltage PEF is applied to the cell membrane, the charges on both sides of the membrane accumulate, thus making the potential difference increase, typically 0.5 to 1.5 V. and when the applied voltage is in excess of a certain value (1V), the cell membrane ruptures and a pore is formed. Below the critical value, however, the pre-pulse transmembrane potential can be restored, and the cell’s metabolic function prior to electroporation can be restored and survived, which is known as the reversible electroporation phenomenon [58,59]. If the potential difference across the cell membrane greatly exceeds a critical value, the cell is unable to recover from the loss of dynamic equilibrium after resealing, which then leads to cellular damage as well as the expulsion of cellular contents, and the cell consequently dies, a phenomenon known as irreversible electroporation [60]. In the literature, the application of PEF in foodstuffs mainly focuses on irreversible electroporation, e.g., pasteurization, dehydration, etc. The application of PEF in freeze drying can improve the quality attributes of fruits and vegetables, reduce the degradation of biologically active compounds, reduce the drying time, etc. Its application in the drying of fruits and vegetables prior to the irreversible electroporation, and its application of the reversible electroporation phenomenon in the drying of fruits and vegetables before the application of the irreversible electroporation phenomenon, have not yet been studied in-depth. The application of reversible electroporation phenomenon before drying has also not been studied in depth yet [61,62].

### 3.4. Microwave

Figure 3d shows a schematic diagram of a microwave drying application. Microwave is an electromagnetic wave with a frequency between 300 MHz and 300 GHz [63]. Microwave frequencies are located between the lower frequencies of radio and the higher frequencies of infrared and visible light. The frequencies commonly used in industry are 915 and 2450 MHz. Waves in the 915 MHz–3 GHz range are readily absorbed by water, fat and sugar in food. In this frequency range, a fluctuating motion is produced, which accelerates polar molecules. This motion is reversed 2450 times in 1 s. This is the operating frequency stated to be used in microwave systems [64]. Microwave freeze drying is a combination of microwave vacuum drying and FD, is the traditional FD method of microwave instead of the traditional heating plate, the basic principle is basically similar to FD, are in the category of sublimation drying. For conventional heating equipment, microwave heating does not require gradient heat from the outer layer for layer-by-layer heating. The water in the foodstuff acts as a dipole and the energy from the microwave is absorbed into the water molecules. Moreover, microwaves can penetrate the depth range of the sample to achieve energy conversion and achieve the effect of volume heating, causing sublimation throughout the sample [65,66,67]. Thus, the shortcomings of the traditional heating method of slow drying are solved, the moisture transfer is faster and the quality of the dried product is higher. The interaction of the fruits and vegetables with the microwaves leads to energy loss and an increase in the elemental temperature, which can be attributed to two phenomena: ion polarization, which is associated with the movement of ions under an alternating electric field and their collision with each other, and dipole rotation, which is associated with the rotation of water molecules under an alternating electric field and the generation of frictional heat [68].

### 3.5. Ultra-High Pressure

Ultra-high pressure (UHP) is to put the food material into liquid medium (such as water, glycerin, etc.), and pressurize the food under a certain temperature using a pressure range of 100 to 1000 MPa, which is applied to the preservation and treatment of food related to the food industry [69]. In food in the ultra-high pressure treatment, the pressure is rapidly transferred to all parts of the food; the ultra-high pressure treatment will destroy the non-covalent bonds in the food over a period of time, so that the microorganisms therein die, cause protein modification, etc., and it will destroy the structure of the cell walls, so that the cell is more permeable, which leads to significant changes in the organizational structure, so that the rate of mass transfer of the product in the dewatering process is increased [70].

## 4. High-Efficiency Physical Fields in Freeze Drying of Fruits and Vegetables

### 4.1. Ultrasound-Assisted Atmospheric Pressure Freeze Drying (AFD)

Atmospheric freeze drying is the sublimation of ice at atmospheric pressure, where a water vapor pressure difference is created between the product and the air, which is the driving force that promotes ice sublimation [71]. Atmospheric freeze drying is an alternative technology to vacuum freeze drying due to its lower cost and continuous drying process [72]. Atmospheric pressure freeze drying has a long drying process due to the low vapor diffusion rate at atmospheric pressure and the blocked internal heat mass transfer, which limits its practical application [71]. Therefore, certain methods to enhance this process are necessary. The auxiliary treatment of ultrasound can improve this disadvantage and the application of ultrasound in the atmospheric pressure freeze drying of fruits and vegetables has been explored by researchers in recent years, and good results have been achieved in this area.

Table 1 summarizes the application and effectiveness of ultrasound in the atmospheric freeze drying of fruits and vegetables. Carrión et al. [73] treated mushroom slices (6 mm thick) with ultrasound at 24.6 kW/m^3^ and 12.3 kW/m^3^ at 21.9 kHz. With the application of ultrasound, the drying kinetics of shiitake mushrooms have been improved to a large extent, reducing drying time by 74% and increasing the effective diffusion coefficient by 280%, without a major impact on sample quality. Moreno et al. [74] subjected two geometries of apples to an ultrasonic atmospheric pressure freeze drying process to explore what effects the application of ultrasound had on the freeze drying process and antioxidation. It was observed that the application of ultrasound increased the drying rate, with the higher the power of the ultrasound, the faster the drying. Ultrasonic drying time was reduced (88% for cylindrical samples and 92% for plate samples) and total energy consumption for drying was reduced by 68.8% (for cylindrical samples) and 78.8% (for plates). For oxidation resistance, there was a degree of reduction in the drying process, which may be related to damage under acoustic stress and oxygen transfer. Garcia-Perez et al. [75] explored what effects ultrasound has on drying at low temperatures and ultimately found a significant reduction in drying time of 41% for applied power. The mass transfer coefficient, as well as the effective moisture diffusivity, increased by 96–177% and 407–428%, respectively, mainly related to mechanical effects, with ultrasound producing alternating compression and expansion at the gas–solid interface in the material being dried. Ultrasound had an obvious influence on the effectiveness of atmospheric pressure freeze drying of apples (30 × 30 × 10 mm), with an increase in ultrasonic power leading to shorter drying times and an effective diffusion coefficient (higher than 4.8 times) at a minimum power of 10.3 kW/m^3^ [76]. Mello et al. [77] carried out the action of ultrasound for atmospheric pressure freeze drying and medium temperature convection drying, respectively, on orange peel. Ultrasound is even more beneficial for atmospheric pressure freeze drying. The application of ultrasound speeds up the drying process and reduces the drying time by approximately 57%, but it still takes a considerable amount of time to achieve the desired final moisture content. Colucci et al. [30] researched the influence of different power of ultrasound on the antioxidant properties of eggplant (8.8 mm and 17.6 mm cube side), using 0, 25 and 50 W at 21.9 kHz for drying experiments, using ascorbic acid content, total phenolic content and antioxidant capacity of the dried product as quality indicators, and found no significant differences compared to samples dried without ultrasound, and that the potential enhancement of the antioxidant response by ultrasound was not significant.

### 4.2. Ultrasound Pretreatment-Assisted Freeze Drying

The effects of ultrasonic pre-treatment combined with freeze drying operations on different quality attributes of fruits and vegetables in the same pattern as auxiliary drying are presented below. Table 2 shows an evaluation study involving the quality assessment of fruits and vegetables after ultrasound pretreatment-assisted freeze drying. Ren et al. [78] investigated the effects of ultrasound and bleaching on the bioactive substances of onions after freezing and hot-air drying, and found that ultrasound increased the retention of bioactive substances after sample drying, thus improving the antioxidant capacity of onion slices, possibly due to ultrasound disrupting the tissue structure and improving the extraction of phytochemicals. Zhang et al. [79] investigated the effects of ultrasound, UHP and combined applications on freeze-dried strawberries and found that the combined application of ultrasound and UHP had the greatest effect on the drying rate as well as energy consumption of strawberries, probably as a result of the formation of microchannels by ultrasound treatment, which increased the mass transfer rate. Figure 4 shows a scanning electron microscope image of the control sample and the treated strawberry sample, and it was clearly evident that the ultrasound sample has well-defined cell walls and enlarged pore structures. For strawberries, Xu et al. [80] reported the effect of different frequency ultrasound treatments on the quality and moisture migration of vacuum freeze-dried strawberry slices. They found that freeze-dried strawberry slices treated with ultrasound reduced drying time by 15.25% to 50%, compared to those without ultrasound treatment. In addition, the effect of dual-frequency ultrasound differed from that of single-frequency ultrasound on strawberries, with dual-frequency ultrasound reducing drying time more than single-frequency ultrasound. The sequential dual-frequency mode ultrasound was an effective pretreatment method for freeze-dried strawberry samples, with quality indicators such as rehydration, color and oxidation resistance significantly higher in the treated samples than in the control. Wang et al. [69] reported that vacuum freeze-dried strawberry slices pretreated with ultrasound had better aroma and taste, with an increase of 19.42% and 23.51% in glucose and fructose content (sweetness) and a decrease of 5.03% in malic acid content (acidity) in the samples, and that the ultrasound pretreatment helped to improve the flavor of the slices. Xu et al. [81] found that ultrasound processing significantly improved drying efficiency, and that the treated okra retained most of its quality characteristics (color, flavor hardness, total phenols and antioxidant capacity). Lyu et al. [82] demonstrated that ultrasound combined with ascorbic acid and CaCl_2_ (UAA) treatment effectively increased the a* (red, green) value, color and total carotenoid content (TCC) of carrots, ultimately by 29.66%, 16.59% and 13.40%, respectively, and improved the thermal stability by 3.40%, and that this treatment facilitated carrot storage. In addition, Lyu et al. [83] performed research in terms of physical structure and carotenoid substances. The UAA treatment resulted in an intact cell structure, reduced porosity and induced carotenoid distribution, not only increasing the total carotenoid content (36.38%) and retention of carotenoids (β-carotene) (51.73%), but also maintaining the high Raman intensity (9986 A.U) of the C=C plane extension and inducing the formation of pigmented carotenoid derivatives. Cao et al. [84] investigated the effect of ultrasonic treatment (UT) of different powers on the quality parameters of freeze-dried barley grass and found that UT (45 W/L, watt per liter) reduced drying time by 14% and energy consumption by 19% and UT (60 W/L) reduced the total number of microorganisms by 33%. The dry samples treated with ultrasound (30 W/L) contained 9.2 g/kg of flavonoids and 10.5 g/kg of chlorophyll, with the highest L* (51.5%) and lowest a* (9.3 g/kg) treated with ultrasound power (10 W/L). At the same time, UT results in a higher glass transition temperature, lower water activity and improved flavor of the dried product.

From these conclusions, it can be seen that ultrasonic pretreatment plays a positive impact during the freeze drying process, shortening the drying time and accelerating the drying speed. It also has influence on the quality parameters of the dried fruit and vegetable products. Ultrasound is a good alternative to drying kinetics for improving the flavor of fruits and vegetables in freeze drying. The main reason for the beneficial effects on drying is the physical effect of cavitation and sponging produced by ultrasound, which alters the cellular microstructure of fruits and vegetables, resulting in cell wall rupture, the formation of microscopic channels, increased porous organization and increased heat mass transfer. However, different ultrasound parameters such as temperature, frequency, power, probe length, medium and simultaneous or continuous application can affect the drying process using ultrasound as a pretreatment.

### 4.3. Infrared Freeze Drying

Table 3 summaries the effect of infrared freeze drying on the drying kinetics of fruits and vegetables. As can be seen from Table 4, an increase in drying kinetics can always be observed when IR radiation is combined with freeze drying. For example, Hnin et al. [85] used infrared freeze drying (IRFD) technology for the treatment of edible rose flowers and proposed an infrared pulsed spray freeze drying technique to evaluate its drying kinetics. They found that infrared pulse-spouted freeze drying (IRPSFD) reduced processing times by 8–30%, and total energy savings by 15–36% compared to FD. Khampakool et al. [86] conducted the effect of infrared freeze drying on banana drying, and found that continuous infrared assisted freeze drying (IRAFD) treatment clearly improved banana drying efficiency and saved more than 70% of time compared to FD, in addition to saving 8.4 × 10^3^ kJ of electricity with continuous IRAFD (2.7 kW/m^2^). Antal et al. [87] investigated the effect of three drying methods, mid-infrared freeze drying (MIR-FD), mid-infrared (MIR) and freezing on the drying characteristics of pears, where mid-infrared freeze drying had the fastest drying rate, saving 14.3 to 42.9% compared to FD, and had better rehydration capacity. Kang et al. [88] tested its effectiveness in drying strawberries under different IRAFD treatment conditions. The results showed that continuous IRAFD not only reduced the power consumption by 42% compared to conventional FD, but also reduced the drying time from 691 ± 19 min to 309 ± 32 min. The drying kinetics were analyzed with different drying models, finding that the Page model fitted the experimental drying curve best.

In addition, while IR freeze drying improves the drying kinetics of the samples, it also has a different effect on the quality of the samples and their active ingredients. For example, Wu et al. [89] developed a new infrared freezing device and reported the effect of freezing infrared drying and conventional freeze drying on the quality of *Cordyceps sinensis* at different drying temperatures. The retention of cordycepin, total phenolic content, hydroxyl radical scavenging activity, reducing ability, 3-octanone, 3-octanol and 1,3-octadiene were all low at 70 °C. Too high a drying temperature adversely affected the quality of the samples, and there was no major difference in the quality of the dried samples compared to TFD. Similarly, Hnin et al. [85] reported similar results for the drying of roses, which were treated at different temperatures using infrared freeze drying, and developed an infrared pulse spray freeze drying technique, which found that the retention of total phenol and anthocyanin content decreased with increasing temperature, probably due to the increased temperature favoring the breakdown of total organic carbon and enhanced anthocyanin degradation. Gu et al. [90] investigated the effects of hot-air drying (HAD), freeze drying, and catalytic infrared combined hot-air drying (CIRD-HAD) and freeze drying (CIRD-FD) on the quality of chives, and found that catalytic infrared freeze drying better preserved the nutrients in the dried samples, including chlorophyll, allicin, vitamin C and volatile components. Chao et al. [91] used FD and freeze drying combined with far infrared drying (FD-FIRD) techniques for drying pumpkin, where the combined drying of FD and FIRD had a greater effect on the physical properties and bioactive composition of the samples. The free phenol content of FD-FIRD treated samples was 14.97–26.60% higher in comparison to FD with increasing time. In particular, the FD for 25 h followed by FIRD for 2 h (FD25-FIRD2) sample showed a 32.23% increase in p-coumaric acid content. In addition, the retention of carotenoids by FD-FIRD was 3.00–3.39 times higher than that of FIRD.

**Table 3 foods-12-04321-t003:** Effects of infrared radiation on drying kinetics.

FVs	The Initial Moisture Content	Drying Conditions	Main Results	References
Apple	5.89 ± 0.1 kg water/kg dm	Voltage: 220 V, wavelength: 3 μm, radiative heat flux: 0.11 W/m^2^	The combination of IR drying and plasma treatment reduced the drying time by 18.0%, 13.0% and 10.5% for 5 mm, 7 mm and 10 mm apple slices, respectively, and reduced the total specific energy consumption.	Khudyakov et al. [92]
Açai puree	94.65 ± 0.10%	Radiation power: 100 W, Infrared intensity: 2.0 W/m^2^, vacuum pressure: 99.8 Pa, T: 32–35 °C	Near IRFD and far IRFD saves 49.42% and 33.40% of drying time, respectively.	Oliveira et al. [52]
Banana	-	Radiant energy: 2.7 kW/m^2^, vacuum pressure: 267 Pa, distance: 150 mm; cold trap temperature: −100 °C	Continuous IRAFD can significantly reduce the drying time up to 213 min, saving more than 70% of time. Continuous IRAFD-2.7 kW/m^2^ fast drying can save electricity up to 8.4 × 103 kJ.	Khampakool et al. [86]
Chives	90.74 ± 0.86%	T = 70 ± 5 °C	CIRD-FD significantly reduces the drying time for each intermediate moisture content by 1.1 to 3.8 h.	Gu et al. [90]
Cordyceps militaris	636.92% db	Pressure: 80 Pa, the heat flux value: 0.703 W/cm^2^, T = 40, 50, 60, 70 °C	Under constant temperature drying conditions, RFD drying time is 7.21–17.78% shorter and energy consumption is 11.88–18.37% lower than TFD.	Wu et al. [89]
Pumpkin	89.91 ± 0.29 g/ 100 g wb	Wavelength: 5–15 μm, T: 60 °C, t: 6 h	Compared to FD, FD-FIRD reduces the drying time and increases the hardness of the dried sample.	Chao et al. [91]
Pear	81.03% wb	Infrared intensity: 3–5.5 kW/m^2^, wavelength of radiation: 2.4–3.0 µm	Mid-infrared freeze drying (MIR-FD) is fast and the drying time is 14.3~42.9% shorter than the FD method, and has a better rehydration capacity.	Antal et al. [87]
Quinoa	10.3 ± 0.05% wb	Maximum heat flux density: 0.703 W/m^2^, W = 3 μm, pressure: 80 Pa, Infrared heating temperature: 50 °C, cold trap temperature: −40 °C	IRFD technology saves 18.2–22.7% of drying time compared to FD technology and maintains good rehydration capacity, texture and color.	Chen et al. [93]
Rose flowers	82%	Pressure: 80 ± 3 Pa, the infrared glass lamps: 100 W, Infrared heating temperature: 50, 60, 70 °C, cold trap temperature: −40 °C	Compared to FD, IRPSFD offers a reduction in processing time of 8–30% and total energy savings of 15–36%.	Hnin et al. [85]
Strawberry	-	Radiation energy: 1.6 kW/m^2^, vacuum: 6.67 Pa; T = 25 °C	Compared to FD, continuous IRAFD significantly reduces drying times and consumes 42% less power.	Kang et al. [88]
Shiitake mushroom	84 ± 1.7%	Wavelength zone: 2.3–3 μm, maximum power: 2.1 kW, Heating intensity: 5.8 kW/m^2^, T = 60 °C	MIRD in combination with FD saves 48% of drying time.	Wang et al. [94]

### 4.4. Pulsed Electric Field Assisted Freeze Drying

Several authors have reported on the use of PEF as a method of pre-treatment for freeze drying fruits and vegetables. Lammerskitten et al. [95] used a pulsed electric field to treat strawberries and bell peppers (a diameter of 3.0 ± 0.1 cm and a thickness of 840.6 ± 0.2 cm) at an electric field strength of 1.0 kV/cm, energy of 0.3 to 6.0 kJ/kg and treatment time of 2.0 to 28.6 ms. As a result of the application of PEF prior to freezing, which reduced the formation of stomata, the degree of shrinkage of strawberries and bell peppers was reduced and the rehydration capacity was increased by 50%, in addition to slow freezing and greater ice crystal formation, increased cell disintegration and mass transfer further reduced volume loss by 50% and 30%, respectively, and hardness by 60%. Similarly, PEF improved the final quality of freeze-dried red beets and pineapples. Ammelt et al. [59] chosen pulsed electric fields as a method for pretreating red beets (10 × 10 × 10 mm) and pineapples (15 × 15 × 15 mm) as freeze-dried under conditions of electric field strength of 1.07 kV/cm and specific energies of 1 kJ/kg and 4 kJ/kg, and found that the treated red beets and pineapples had less shrinkage, more uniform shape and better visual quality. There was no significant difference in the reduction of drying time. Liu et al. [96] employed a combinative use of PEF and vacuum-dried potatoes (25 mm in diameter and 2.5 mm in thickness), which had a positive effect on the freezing process of the samples and accelerated the freeze drying process. PEF is more commonly used in freeze drying apples. Lammerskitten et al. [97] investigated the effect of PEF on the kinetics and physical properties of freeze drying apple (72 ± 3 mm) tissue. The application of PEF reduced the drying time by 57%, and the drying rate of treated apples was approximately twice that of untreated apples, probably due to the reduced internal resistance to heat and mass transfer resulting in faster drying. In addition, PEF induced electro-permeability of the cell membrane, enhancing the diffusion process and increasing the effective diffusion coefficient by 44%. Lammerskitten et al. [98] treated apples (72 ± 3 mm) with PEF prior to freeze drying at 1.07 kV/cm and specific energies of 0.5, 1 and 5 kJ/kg. It was found that the treated samples could retain their macroscopic shape well, inhibit and shrink the progression of large pores and have a higher brittleness. Figure 5 shows macroscopic and scanning electron microscopy photographs of dried apples before and after the PEF treatment. Similarly, Parniakov et al. [36] investigated the effect of pulsed electric field treatment on vacuum freeze drying of apples (2.9 mm in diameter and 5 mm in thickness) at an electric field strength of E = 800V/cm. Excellent macroscopic shape and a large rehydration capacity were observed in PEF-treated apples, with an 86-fold increase in the pore size of the samples and a high level of rehydration capacity (≈1.3) observed at disintegration index Z = 0.96. Figure 6 shows macroscopic and microscopic photographs of untreated and PEF-treated apple samples after vacuum freeze drying. Faster cooling and drying processes were observed after the pulsed electric field treatment in terms of drying rates. Tylewicz et al. [99] treated apples (d = 20 mm, height = 20 mm, m = 2.6 g) at a frequency of 3 Hz and electric field strengths of 0.3, 0.6, 0.9 and 1.2 kV/cm, and found that the cell disintegration index increased with increasing electric field strength, number of pulses and energy delivered to the sample. PEF treatment affected the structural integrity and continuity of the cells (increased ZP index), as shown by the redistribution of water between the different compartments. Nowak et al. [100] treated the apples (10 mm thick) with a pulsed electric field with a pulse count of 0 to 160 kJ/g and an energy range of 0 to 1327 kJ/g. The outcome showed that PEF had a positive effect on the drying rate, but the PEF pretreatment damaged the cells to a degree of 25% (conductivity disintegration index Z = 0.25).

In summary, PEF has been reported as a pretreatment method for a variety of foodstuffs prior to drying with successful results. PEF-assisted processing has wide-ranging prospects in fruits and vegetables processing. The application of PEF improves drying kinetics, increases drying rates and effective diffusion coefficients while also reducing disruption of the product’s cell structure from freeze drying and maintaining macroscopic shape. The primary reason for this phenomenon is still the electroporation principle of PEF, which disrupts the integrity of the cell membrane, increases its permeability and facilitates the heat transfer process [101].

**Table 4 foods-12-04321-t004:** Application of pulsed electric field assisted freeze drying of fruits and vegetables.

FVs	The Initial Moisture Content	Drying Conditions	Main Results	References
Apple	85.1%	Electric field strength: 1.07 kV/cm, specific energy inputs: 0.5 kJ/kg, 1 kJ/kg	The PEF treatment reduced the apple drying time by 57% and enhanced the diffusion process, increasing the effective diffusion coefficient by 44%.	Lammerskitten et al. [97]
Apple	83 ± 1%	Electric field strength: 1.07 kV/cm, specific energy inputs: 0.5 kJ/kg,1 kJ/kg, 5 kJ/kg	The treated samples retained their macroscopic shape better, inhibited and contracted macroporous processes, and had a high degree of brittleness.	Lammerskitten et al. [98]
Apple	85% wb	Electric field strength: 800 V/cm	PEF-treated apples had excellent macroscopic shape and greater rehydration capacity, with an 86-fold increase in the pore size of the samples and a high level of rehydration capacity.	Parniakov et al. [36]
Apple	-	Electric field intensity: 1 kV/cm, voltage: 24 kV	PEF had a positive effect on drying rate, but PEF pretreatment damaged the cells by 25%.	Nowak et al. [100]
Potatoes	89.91 ± 0.29 g/ 100 g wb	Electric field strength: E = 600 V/cm, total treatment time 26 of tPEF = 0.1	PEF treatment had a positive effect on the freezing process of potatoes, and it accelerated the drying process.	Liu et al. [102]
Red beets and pineapples	85.76 ± 1.31%, 82.36 ± 1.27%	Electric field strength: 1.07 kV/cm, specific energy inputs: 1 kJ/kg, 4 kJ/kg	After PEF treatment, the samples have less shrinkage, more uniform shape and better visual quality.	Ammelt et al. [59]
Strawberries and bell peppers	-	Electric field strength: 1.0 kV/cm, specific energy inputs: 0.3–6 kJ/kg	This resulted in a 50% increase in rehydration capacity, a 50% and 30% reduction in volume loss, respectively, and a 60% increase in hardness.	Fauster et al. [103]
Strawberries	90 ± 0.8%	Electric field intensity: 1.07 kV/cm, specific energy input: 1 kJ/kg	Pretreated strawberries have better shape and volume retention and better visual quality, and are structurally dense.	Lammerskitten et al. [95]

### 4.5. Microwave Freeze Drying (MFD)

Table 5 summarizes research papers on the impact of infrared freeze drying on the drying kinetics of fruits and vegetables. Overall, microwave-assisted freeze drying reduced the drying time in comparison with FD; for example, Xu. et al. [104] studied the drying effect of microwave freeze drying, atmospheric pressure freeze drying and freeze drying on mushrooms after drying, and the three methods had different effects upon the drying time and quality of mushrooms. Compared to FD and AFD, MFD resulted in better drying efficiency with acceptable quality, while AFD treated mushrooms had the longest drying time but the lowest energy consumption. Duan et al. [105] used FD, MFD, heat pump drying (HPD) and AFD to dry hawthorn. The results showed that compared with FD and AFD, MFD saved drying time and had high acceptability of hawthorn products, but it took longer time and consumed more energy than HPD. Therefore, MFD can replace the traditional FD to produce high-quality hawthorn products. However, different results were observed compared to other combined methods. Yan et al. [106] used the three methods of microwave-assisted vacuum drying (MWVD), microwave-assisted freeze drying (MWFD) and microwave-enhanced spray bed drying (MWSD) to treat carrots, and studied the differences between the three methods on carrots in terms of drying rate, drying uniformity, rehydration rate and energy consumption, etc. The results showed that the drying rates of MWVD and MWSD are both higher than MWFD, and MWSD has the highest drying rate (3.5 W g^−1^), but the rehydration rate of MWFD carrot slices was comparable to that of the FD product and better than the other two treatments. Liu et al. [107] also investigated the drying effect of MWFD, MWVD and microwave-assisted spouted bed drying (MWSBD) on purple-fleshed sweet potatoes. The drying times of the three methods differed, with MWSBD taking the shortest time to complete drying and MWFD taking the longest as well as consuming the most energy; about twice as much as MWVD.

Many studies have evaluated the effect of microwave-assisted freeze drying on the quality of dried samples and the damage to the quality of the product after drying is one of the criteria for selecting a drying method. The latest research on the quality of MFD-dried vegetables and fruits is shown in Table 6. Duan et al. [108] demonstrated the impact of microwave freeze drying and microwave vacuum drying on the physicochemical properties, microstructure, etc., of the products obtained after yam drying. The various drying methods had a noticeable effect on all aspects of protein content, starch and color of the yam powder, with MFD retaining a good color similar to FD, probably due to the low temperature vacuum environment resisting the occurrence of the Maillard reaction and oxidation of ascorbic acid. A number of studies have shown that the quality of fruit and vegetables does not change significantly when compared with conventional freeze drying; for instance, Li et al. [109] used a combination of freeze drying and microwave assisted vacuum drying to dehydrate apples, which showed that the nutritional value of the combined apples was similar to the process used alone.

**Table 5 foods-12-04321-t005:** Effect of microwave freeze drying on the drying kinetics of fruits and vegetables.

FVs	The Initial Moisture Content	Drying Parameters	Main Results	References
Apple	-	Microwave power: 3.18 W/g, Vacuum pressure: 9.15 kPa	FD-VMD combination drying reduces the total processing time of apples by up to 40%.	Li et al. [109]
Banana	3.25 ± 0.03 g^−1^ db	Microwave power: 2, 2.5, 3, W/g; Vacuum pressure: 0.1 kPa, microwave frequency: 2450 MHz	The drying time decreases with increasing microwave power and the rehydration rate of the MFD sample is better than the MVD sample at >4.5.	Jiang et al. [110]
Banana	3.25~3.67 g^−1^ db	Microwave power: 2 W/g, Vacuum pressure: 0.1 kPa, microwave frequency: 2450 MHz	Compared to FD, MFD can reduce drying time by 50% and has better rehydration capacity.	Jiang et al. [111]
Carrot	-	Microwave power: 2 W/g, Vacuum pressure: 0.1 kPa, cold trap temperature: 40 °C, microwave frequency: 2450 MHz	The rehydration rate of MWFD carrot slices was comparable to that of the freeze-dried (FD) samples and better than that of the MWVD and MWSD samples, but the drying rate was lower than that of the MWVD and MWSD.	Yan et al. [106]
Cordyceps militaris	-	Microwave power: 660, 760, 860 W, absolute pressure: 0.08 kPa, cold trap temperature: −40 °C	The final moisture content, water activity, rehydration time and rehydration rate of the MPFFD samples were similar to those of the FD samples, but MPFFD reduced the total drying time by 71.9%.	Wu et al. [112]
Hawthorns	73.68 ± 0.15% wb	Microwave power: 1 kW, Vacuum pressure: 0.05 Pa, cold trap temperature: −40 °C	MFDs are more energy intensive and take longer to dry than HPDs.	Duan et al. [105]
Lettuce	-	Microwave power: 800 W, microwave frequency: 2450 MHz	MFD samples take more than 20% longer than PSMFD in total drying time and PSMFD dries more uniformly.	Wang et al. [113]
Mushrooms	90% wb	Microwave power: 1 kW, Vacuum pressure: 0.05 kPa, cold trap temperature: −40 °C	The drying time for MFD is 8 h, which is much lower than AFD and FD.	Xu. et al. [104]
Purple-fleshed sweet potato granules	-	Microwave power: 4 W/g, Vacuum pressure: 4.5 kPa	MWFD takes longer than MWVD and MWSBD to complete drying and MWFD consumes approximately twice as much energy as MWVD.	Liu et al. [107]
Pineapple	-	Microwave power: 6 W/g, microwave frequency: 2450 MHz	Microwave-assisted freeze drying is 34.5% more energy efficient than pure vacuum drying and reduces drying time by 33.3%.	Chen et al. [114]

**Table 6 foods-12-04321-t006:** Effect of microwave freeze drying on quality parameters of fruits and vegetables after drying.

FVs	The Initial Moisture Content	Drying Conditions	Main Results	References
Banana	3.34 ± 0.03 g/g db	Microwave power: 2 W/g, Vacuum pressure: 0.1 kPa, cold trap temperature: 40 °C, microwave frequency: 2450 MHz	Changes in starch content, reducing sugar content, structure and color of the banana slices are greatest in the primary drying stage, and the greatest change in swelling rate is in the secondary drying stage.	Jiang et al. [115]
Carrot	-	Microwave power: 2 W/g, Vacuum pressure: 0.1 kPa, cold trap temperature: 40 °C	MWFD carrot slices have the highest retention of carotenoids and vitamin C.	Yan et al. [106]
Lettuce	-	Microwave power: 800 W, microwave frequency: 2450 MHz	Compared to MFD, PSMFD samples have a uniform and dense microstructure, less color discoloration and better quality.	Wang et al. [113]
Purple-fleshed sweet potato granules	-	Absolute pressure: 10–30 kPa	MWFD was able to maintain a high phalloidin content of 74.98% compared to 71.41% and 53.76% for MWVD and MWSBD, respectively. The MWFD product also had good brittleness but the MWSBD product had better color and appearance.	Liu et al. [107]
Yam	79.25 ± 1.84 g/100 g	Microwave power: 2 W/g; Vacuum pressure: 0.08 kPa	The MFD treated samples maintained good color and had the highest PV and FV values.	Duan et al. [108]

The MFD has certain advantages in fruit and vegetable drying, which can save drying time and reduce energy consumption and can replace traditional FD, but it is important to control the drying parameters to ensure drying efficiency as well as sample quality.

## 5. Energy Consumption

Drying process should be considered not only product quality, we also need to pay more attention to the economic cost, and the biggest obstacle to limit the promotion and use of new drying technology may be the energy consumption and cost problems [116]. Therefore, to maintain the appearance and quality of the product, while reducing the time and energy consumption of the freeze drying process, has become the most important concern in the production of freeze drying [54].

Freeze drying equipment has more components and is expensive, and the long time required for the whole process leads to high energy consumption and higher costs, so it limits its application to some extent. For example, 1000 kg of green onions consume 1080 kWh of electricity after freeze drying [117]. Kaveh et al. [118] compared green peas after treatment using seven drying methods, and found freeze drying to be the most expensive drying method. In recent years, researchers have demonstrated the utility of microwave- and infrared-assisted freeze drying in reducing energy consumption, but ultrasound and pulsed electric fields have been less well studied in terms of energy consumption. Chen et al. [114] discussed the effect of microwave-assisted freeze drying on pineapple quality and energy consumption after treating pineapple with microwave-assisted freeze drying. Under the conditions of transition moisture content of 20%, drying temperature of 40 °C, rotary speed of 8 rpm, and microwave power density of 6 W/g, microwave-assisted freeze drying saves 34.5% of energy and shortens the drying time by 33.3%, compared with pure vacuum drying. Jiang et al. [119] found that treating banana chips with microwave freeze drying can save up to 35.7% of energy and 40% of drying time compared to freeze drying, and that increasing the heating power in the secondary drying stage reduces energy consumption and shortens drying time. Microwaves are an effective way to reduce drying costs. After Cordyceps sinensis was processed by infrared freeze drying and freeze drying, Wu et al. [89] found that both of them could effectively reduce the drying time and energy consumption under the premise of maintaining the same quality of samples. Hnin et al. [85] treated edible moonflowers with infrared freeze drying technology and conventional freeze drying separately to study the effects of both on their product quality and energy consumption, and found that IRPSFD reduced the processing time by 8–30% and saved 15–36% of the total energy consumption compared with FD. Khampakool et al. [86] investigated the application of infrared-assisted freeze drying to banana snacks and found that FD had the highest total power consumption, while continuous IRAFD-2.7 kW/m^2^ rapid drying could save power up to 8.4 × 10^3^ kJ.

Most of the literature is oriented towards increasing the rate of drying, with little consideration of the economic cost of purchasing capital for the equipment. The most significant cost of a freeze dryer is the initial cost, which is the investment in raw materials and equipment, rather than the utilization of the freeze dryer, whereas the operating cost is relatively low. The application of physical fields in freeze drying reduces the drying time of food products, increases the drying rate, improves the productivity and reduces the operating cost, which is higher per unit of production of a laboratory-scale freeze dryer as compared to a freeze dryer in industry.

## 6. Trends and Challenges for the Future

In this context, we would like to suggest the establishment of the following developments to provide some direction for the future of freeze drying of fruit and vegetables.

### 6.1. Establishing the Correlation between Drying Methods and Fruit and Vegetable Characteristics

Various highly efficient physical field drying technologies have their own advantages and disadvantages in fruit and vegetable dehydration, and can solve different problems for different types of fruits and vegetables, producing different effects on different fruits and vegetables. This requires researchers to properly understand their characteristics, to match different fruits and vegetables with the appropriate auxiliary freeze drying technology and processing conditions, and to establish the relevance of the physical technology in the freeze drying process to the dried fruit and vegetable product in order to optimize the dried product to the maximum. For example, microwave and ultrasonic drying is not uniform, and infrared radiation has a more severe loss of color in fruits and vegetables. Finding the right type of dehydration for different raw materials is of great relevance to the preservation and freshness of food [120].

### 6.2. Artificial Intelligence Technology Drives the Development of Fruit and Vegetable Drying

At present, artificial intelligence technology is less used in the food industry. With the rapid development of computers and artificial intelligence, the application of artificial intelligence technology to assist efficient physical fields for fruit and vegetable drying is the future development trend. Artificial intelligence technology can address the shortcomings of efficient physical field technology and better help us understand the drying process. The application of artificial intelligence mainly includes algorithms, neural networks, computer vision, and so on [121]. The use of algorithms allows prediction of drying models, which helps us to master the drying process and drying parameters and save costs [122]. Computer vision is applied to monitor changes in the color and shape of fruits and vegetables during the drying process and to obtain pictures of dried samples in real time [123]. There is also electronic nose technology that can be used to monitor and analyze changes in flavor and volatile compounds in samples during drying. Makarichian et al. [124] used an electronic nose to identify variations and differences in the aroma of garlic when dried in different ways. Zhang et al. [125] used an electronic nose to analyze the pattern of changes in volatile compounds during the drying process of coffee beans. The use of artificial intelligence in fruit and vegetable drying can provide new tools and directions for the development of food products.

### 6.3. Synergistic Application of Multiple Drying Methods

Although progress has been made so far, more research and technical knowledge is needed to expand the industrial application of the process. In addition to the options related to efficient physical fields, more importantly, synergistic effects and the drying characteristics of fruits and vegetables should be studied in depth in the future. In view of the advantages and disadvantages of existing drying aids, the choice of a combination of drying methods is an important research direction to improve the quality and drying rate of dried fruit and vegetable products [126,127], but further research is needed to grasp the transition point between each technique and the optimum control conditions.

### 6.4. Reducing Economic Losses

In addition, all improvement studies must not be carried out on the basis of ignoring the economics; cost reduction is an important area of research in obtaining dried fruit and vegetable products, and excessive economic demands as well as losses will hinder the pace of application in industry [128]. The development of environmentally friendly and economical new drying equipment can improve the added value of dried fruit and vegetable products to maximize production benefits and further promote the development of related industries.

## 7. Conclusions

FD is considered one of the best food preservation processes for foodstuffs, and is broadly used for drying fruit and vegetables. The method is most popular, as it results in a high-quality dried product with an extended storage period. However, this method of preservation is costly, requires expensive equipment and consumes high levels of energy. The combined use of efficient physical field technology and the freeze drying technique has advanced significantly. This review shows that ultrasonic atmospheric pressure freeze drying, ultrasonic pretreatment freeze drying, infrared radiation freeze drying, pulsed electric field freeze drying and microwave-assisted freeze drying can significantly reduce drying time, increase the heat mass transfer process and improve the quality of the final product. Future objectives should be to enhance the FD process through innovative technologies and to overcome the challenges posed by FD.

## Figures and Tables

**Figure 1 foods-12-04321-f001:**
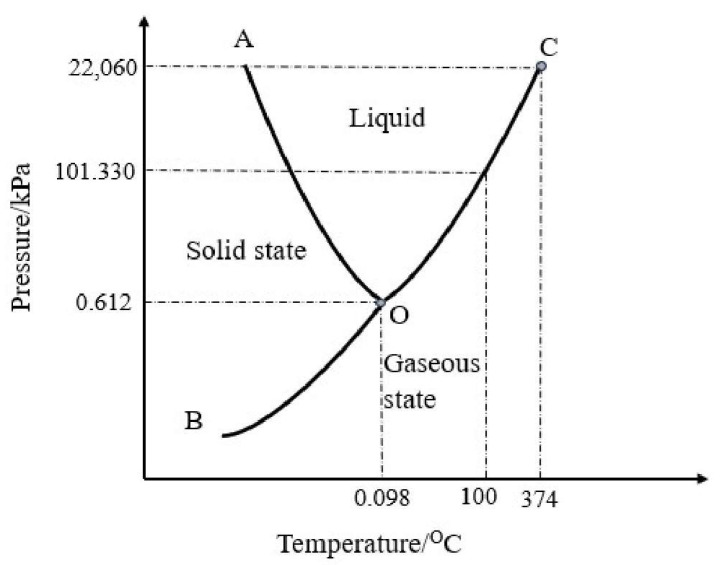
Three phase diagram of water. (O is the three-phase point of water, C is the critical point of water, OA is the boundary between solid and liquid states, and OB is the boundary between solid and gaseous states.).

**Figure 3 foods-12-04321-f003:**
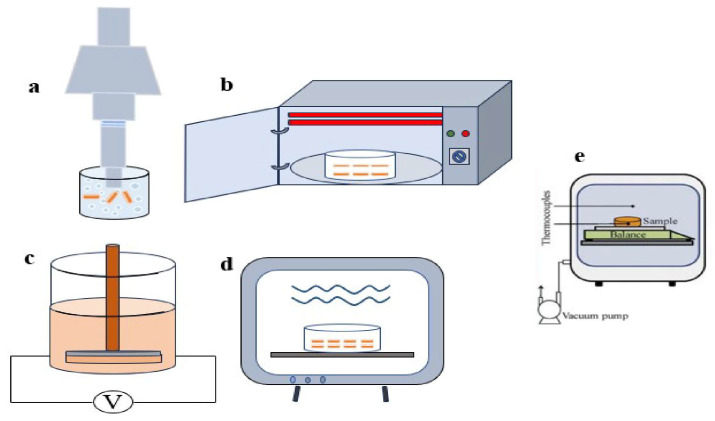
Ultrasound (**a**), infrared radiation (**b**), pulsed electric field (**c**), microwave (**d**), and freeze drying (**e**) [36] sketch. Reproduced with permission from copyright owner; published by Elsevier, 2016.

**Figure 4 foods-12-04321-f004:**
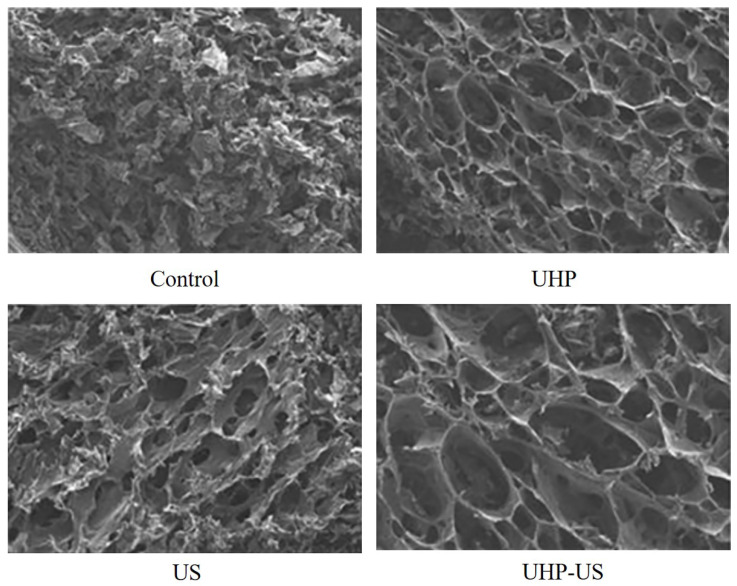
Scanning electron micrographs of vacuum-freeze dried strawberry chips with different pretreatment methods. UHP: ultra-high pressure, US: ultrasound, UHP-US: ultra-high pressure in combination with ultrasound [79]. Reproduced with permission from copyright owner; published by Elsevier, 2020.

**Figure 5 foods-12-04321-f005:**
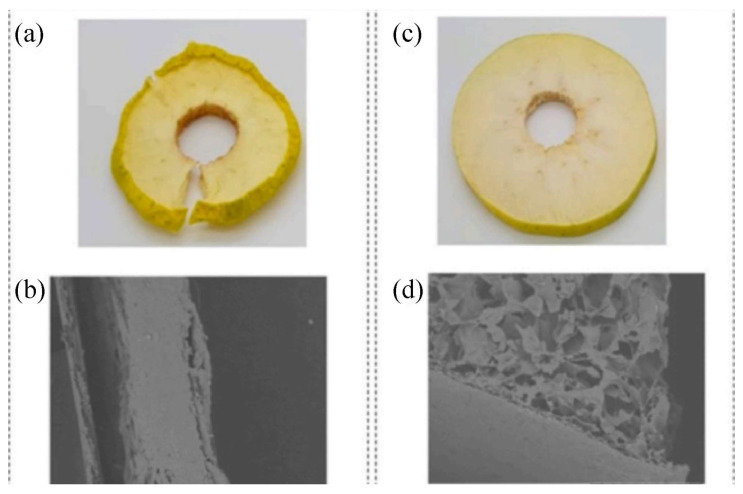
SEM and macroscopic photos for the untreated (**a**,**b**) and PEF-treated (Wspec = 1 kJ/kg, E = 1.07 kV/cm) (**c**,**d**) dried apple tissue. Magnification of ×100 [98]. Reproduced with permission from copyright owner; published by Elsevier, 2019.

**Figure 6 foods-12-04321-f006:**
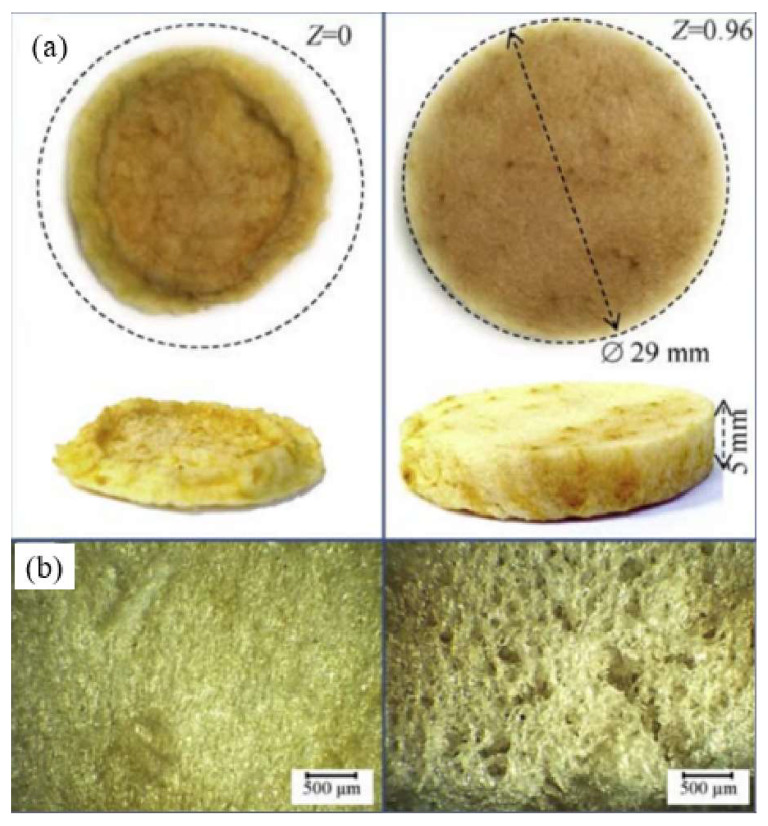
Macroscopic (**a**) and microscopic (**b**) photos of untreated (Z = 0) and PEF pre-treated (Z = 0.96) apple samples after vacuum freeze drying. The initial apple disc-shaped samples before drying had diameter of di = 2.9 mm and thickness of hi = 5 mm [36]. Reproduced with permission from copyright owner; published by Elsevier, 2016.

**Table 1 foods-12-04321-t001:** Application of ultrasound assisted atmospheric pressure freeze drying of fruits and vegetables.

FVs	The Initial Moisture Content	Drying Conditions	Main Results	References
Apple	6.2 ± 0.4 kg water/kg dm	T: −10 °C; Frequency: 21.7 kHz; Ultrasonic power: 0, 10.3, 20.5 and 30.8 kW/m^3^	The application of ultrasound resulted in a reduction in drying time (88% for cylindrical samples and 92% for sheet samples) and a reduction in total drying energy consumption of 68.8% (cylindrical samples) and 78.8% (sheet samples).	Moreno et al. [74]
Apple, carrot and eggplant		T: −10, 0, 10, 20 °C, Air velocity: 1, 2, 4 m/s; Ultrasonic power 0, 10.3, 20.5 kW/m^3^	When ultrasound is applied, the total energy consumption of the process is reduced by 70%.	Merone et al. [72]
Apple	6.2 ± 0.4 water/kg dm	T: −10 °C, 15% RH; Frequency: 21 kHz; Air velocity: 2 m/s; Ultrasonic power: 0, 10.3, 20.5, 30.8 kW/m^3^,	The effective diffusion coefficient of ultrasound is 4.8 times higher at the lowest power (10.3 kW/m^3^), and the higher the power used, the better the drying effect.	Brines et al. [76]
Carrot, eggplant, and apple	7.58 ± 0.90, 6.10 ± 0.37, and 14.57 ± 0.27 water/kg dm	T: −14.1 °C, 7 ± 3% RH; Frequency: 21.9 kHz; Air velocity: 2 ± 1 m/s; Ultrasonic power: 19.5 kW/m^3^	In the presence of ultrasound, the drying time was shortened (about 65–70%), and the mass transfer coefficient and effective moisture diffusion rate were increased by 96–170% and 407–428%, respectively.	Garcia-Perez et al. [75]
Eggplant		Frequency: 21.9 kHz; T: (−5, −7.5, −10 °C); Air velocities (2, 5 m/s); Ultrasonic power: 0, 25, 50 W	No significant difference compared to samples dried without ultrasound, the potential effect of ultrasound to enhance the antioxidant response was not significant.	Colucci et al. [30]
Orange peel	2.47 ± 0.08 kg water/kg dm (dry basis moisture content)	AFD (T: −10 °C, MTD (T: 50); Air velocity: 1 m/s; Ultrasonic power: 20.5 kW/m^3^	In the presence of ultrasound, the freeze drying process is accelerated and the drying time is reduced by about 57% compared to medium temperature convection drying (MTD).	Mello et al. [77]

**Table 2 foods-12-04321-t002:** Application of ultrasound pretreatment assisted freeze drying of fruits and vegetables.

FVs	The Initial Moisture Content	Drying Conditions	Main Results	References
Carrots	-	T: 25 °C; t: 10 min; Frequency: 40 kHz; Ultrasonic power: 100 W.	Ultrasound combined with ascorbic acid and after ultrasonic treatment with ascorbic acid and CaCl_2_ (UAA-CaCl_2_), the a* value, color and TCC of carrots were increased by 29.66%, 16.59% and 13.40%, respectively, and the thermal stability was increased by 3.40%.	Lyu et al. [82]
Carrots	88 g/g wb	T: 25 °C; t: 10 min; Frequency: 40 kHz	The UAA treatment resulted in structural integrity of the cells, reduced porosity, induction of carotenoid distribution and increased retention of carotenoid material (51.73%).	Lyu et al. [83]
Grass	-	T: 10 °C; t: 10 min; Frequency: 20 kHz; Ultrasonic power: 10, 30, 45, 60 W/L	Compared to untreated, ultrasonic power (10 W/L) had the highest L* (51.5%) and lowest a* (9.3%). UT (30W/L) treated dry samples had 9.2 g/kg of flavonoids and 10.5 g/kg of chlorophyll. UT (45 W/L) reduced drying time by 14% and energy consumption The UT (45 W/L) reduced drying time by 14% and energy consumption by 19%. UT (60 W/L) reduced the total number of microorganisms by 33%.	Cao et al. [84]
Onions	-	T: 70 °C; t: (1, 3, 5 min); Frequency: 20 kHz; Ultrasonic power: 24.4–61 μm	The retention of bioactive substances in the samples was increased after freeze drying compared to hot-air drying, thus improving the antioxidant capacity of the onion slices.	Ren et al. [78]
Okra	87.17 ± 0.68% dm	T: 25 °C; t: 30 min; Frequency: 40 kHz; Ultrasonic power: 25 W/L	The application of ultrasound and different freeze–thaw treatments reduced the drying time by 25.0% to 62.50% and the total energy consumption by 24.28% to 62.35%.	Xu et al. [81]
Strawberry	85.62 ± 2.76% dm	T: −20 °C; t: 25 min; Frequency: 40 kHz; Ultrasonic power: 200 W	The combination of ultrasound and high-pressure treatment promotes the diffusion of water, reduces energy consumption and also improves the color and antioxidant activity of the product.	Zhang et al. [79]
Strawberry	92.65 ± 0.83% wb (wet basis moisture content)	T: 25 ± 1 °C, t: 30 min, Frequency: 20, 40 kHz; Ultrasonic power: 30 W/L	The drying time of the ultrasound-treated strawberry slices was reduced by 15.25% to 50.00% compared to the control, with dual-frequency ultrasound reducing drying time even more and providing higher quality.	Xu et al. [80]
Strawberry	-	T: 25 °C; t: 25 min; Frequency: 40 kHz	The aroma and taste of the vacuum freeze-dried strawberry slices pretreated with ultrasound were superior to those of the UHP-treated samples, with the glucose and fructose contents (sweetness) of the samples increasing by 19.42% and 23.51%, respectively, and the malic acid content (acidity) decreasing by 5.03%.	Wang et al. [69]

## Data Availability

Data are contained within the article.

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
