# Peer review of "Novel Efficient Physical Technologies for Enhancing Freeze Drying of Fruits and Vegetables: A Review"

_foods, 2023, doi:10.3390/foods12234321_

Round 1

Reviewer 1 Report

Comments and Suggestions for Authors

The manuscript presented for review is a very interesting study on the possibilities of improving the sublimation drying process. The work is very well written, although there are some shortcomings in it (I list them below).

In my opinion, the paper lacks a chapter on the cost of using novel efficient physical technologies for enhancing freeze drying, and thus whether it is really possible to reduce the cost, by reducing the time, of sublimation drying. I would ask that such a chapter be added in which, in addition, the possibility of applying such assisted drying on an industrial scale is described, with an indication of whether such a solution is in general possible given the cost of purchasing and implementing all the novel efficient physical technologies for enhancing freeze drying described in the manuscript.

The paper contains the following minor errors:
Line 83: the °C is missing .
Figure 1: please add definition of points A, B, C, O in the caption of Figure 1.
Line 120-121: ... "otherwise it will lead to denaturation and an eventual residual of 0.5-3% moisture...." Denaturation of what?
Line 232: ... "In recent years, researchers have combined..." Which researchers? Literature sources are missing.
Table 1: please capitalize or lowercase all phrases in the first column
Line 552: please add a literature source.

Author Response

Responses to Reviewer: 

In my opinion, the paper lacks a chapter on the cost of using novel efficient physical technologies for enhancing freeze drying, and thus whether it is really possible to reduce the cost, by reducing the time, of sublimation drying. I would ask that such a chapter be added in which, in addition, the possibility of applying such assisted drying on an industrial scale is described, with an indication of whether such a solution is in general possible given the cost of purchasing and implementing all the novel efficient physical technologies for enhancing freeze drying described in the manuscript.

Response: Thanks very much for your suggestion. We have added this section (5. Energy consumption) in Line 605-635.

The paper contains the following minor errors:

Line 83: the °C is missing.

Response: Thanks very much for your suggestion. Point C is the critical point for water (374 oC, 22, 060 kPa), freezing of the sample occurs when the temperature falls below 0.01 oC and water sublimates directly when the pressure falls below 0.612 kpa.

Line 120-121: ... "otherwise it will lead to denaturation and an eventual residual of 0.5-3% moisture...." Denaturation of what?

Response: Thanks very much for your suggestion. In addition to this, a higher temperature is required to complete the desorption process, but the temperature should not be too high and needs to be strictly controlled, requir-ing a temperature below the glass transition temperature of the foodstuff, otherwise fruits and vegetable will lead to denaturation and an eventual residual of 0.5-3% moisture [9].

Line 232: ... "In recent years, researchers have combined..." Which researchers? Literature sources are missing.

Response: Thanks very much for your suggestion. We have made additions. In recent years researchers have combined freeze drying and infrared [47-49].

Table 1: please capitalize or lowercase all phrases in the first column

Response: Thanks very much for your suggestion. We have corrected and standardized upper/lower case.

Line 552: please add a literature source

Response: Thanks very much for your suggestion. We've made changes.

We believe that the revisions made based on those comments have significantly improved the manuscript. We look forward to your information about our revised manuscript and thank you again.

Best regards,

Kai Fan

Reviewer 2 Report

Comments and Suggestions for Authors

General comments:

In my opinion, the authors completely ignored the issue of the nutritional value of freeze-dried fruits and vegetables. Please supplement the review with relevant information (e.g., https://doi.org/10.1080/07373937.2019.1599905).

Energy efficiency, sustainability are key aspects for any economy. The study lacks a review of information on the energy efficiency of the combined drying process, especially with ultrasound. The authors should include information on energy requirements in a separate chapter in the study. Such information will significantly increase the value of the publication.

Chapter 5 increases the value of the publication by presenting trends and challenges for the future of freeze drying of fruits and vegetables. However, this chapter is rather concise. Please supplement with relevant literature data (including https://doi.org/10.1016/j.tifs.2020.06.005; https://doi.org/10.3390/pr8111399 and others)

Specific comments:

l.55 - drying of duck egg whites goes beyond the topic of the publication. Delete the passage.

l.60-61 - indicate the source of the information.

l.82 - the information refers to water, not to any sample; please correct.

l.96 - please check use of capital letter.

l.116 - please correct the use of a capital letter.

l.133 - please correct the literature reference - name needed.

l.154-160 - models without any description (symbols?) are useless. Please comment more widely on the models or remove them from the text.

l.154-155 - models (1) and (2) are illegible; please use smaller letters in the indexes.

l.160 - please correct errors in formula (4).

l.167-168 - please revise the content; ultrasound covers a much wider range of frequencies (above 100 kHz) and intensities (below 1 W/cm²).

l.185 - porous structures have high acoustic resistance; please verify the statement in the sentence.

l.221 - please correct the unit notation.

l.233 - first use of the abbreviation FD; please explain.

l.295 - please verify and correct the information on drying time.

Table 1 - please organize alphabetically by first column.

l.319 - please verify use of capital letter.

l.324 - first use of the abbreviation UHP; please clarify.

l.341 - unnecessary period.

l.349 - parameter a* refers to color; please rephrase.

l.335 - revise and correct notation of arbitrary unit.

l.358 - necessary clarification of "W/L" designation (watt per liter?).

l.363 - unnecessary abbreviation in parentheses (Tg); it is used only once.

Table 2 - please arrange alphabetically by first column.

l.404 - the information on reducing time is meaningless, because the authors do not report the initial time. Reword the sentence.

l.409-410 - use italic in Latin name.

l.423 - revise and correct the use of a capital letter.

l.427 - the explanation of the FD25-FIRD2 designation is missing; please clarify.

Table 3 - please organize alphabetically by first column; insert 2 as superscript (Main results - Banana); correct use of capital letter in unit (kW).

l.467 and the next - please standardize the way ofunits of measurement are written; write with "/" or write with exponent of a power.

l.470 - please clarify the abbreviation ZP; it appears only in this line of text.

Figure 7 - the caption on photos (b) and (d) is illegible; include a clear linear scale; please improve the quality or remove the illegible part of the photo.

l.483 - remove "di=" and "hi=".

l.500 - first occurrence of the abbreviation HPD; please clarify.

l.525 - please verify and correct the name of the reaction.

Table 4 - please organize alphabetically by first column; please use capital letters correctly; please verify and standardize notation of units in column 2 (e.g. The initial moisture content - Banana). 

Table 5 - please organize alphabetically by first column.

l.592 - the freeze drying technique has advanced significantly and the authors should verify/correct the theses posed in this sentence.

References - please verify the notation of journal names (abbreviated names / full names)

l.636 - please verify and correct the names of authors in the publication.

l.676 and the next - please verify the notation of the journal title (capital letters).

Has permission been obtained from the authors to use Figures 5,6,7,8 in this publication?

Author Response

Responses to Reviewer: 

In my opinion, the authors completely ignored the issue of the nutritional value of freeze-dried fruits and vegetables. Please supplement the review with relevant information (e.g., https://doi.org/10.1080/07373937.2019.1599905).

Response: Thanks very much for your suggestion. We have added the contents.

Energy efficiency, sustainability are key aspects for any economy. The study lacks a review of information on the energy efficiency of the combined drying process, especially with ultrasound. The authors should include information on energy requirements in a separate chapter in the study. Such information will significantly increase the value of the publication.

Response: Thanks very much for your suggestion. We have added energy efficiency in Line 605-639.

Chapter 5 increases the value of the publication by presenting trends and challenges for the future of freeze drying of fruits and vegetables. However, this chapter is rather concise. Please supplement with relevant literature data (including https://doi.org/10.1016/j.tifs.2020.06.005; https://doi.org/10.3390/pr8111399 and others

Response: Thanks very much for your suggestion. We have added the literature data.

Specific comments:

l.55 - drying of duck egg whites goes beyond the topic of the publication. Delete the passage.

Response: Thanks very much for your suggestion. We have deleted.

l.60-61 - indicate the source of the information.

Response: Thanks very much for your suggestion. We have indicated.

l.82 - the information refers to water, not to any sample; please correct.

Response: Thanks very much for your suggestion. We have corrected.

l.96 - please check use of capital letter.

Response: Thanks very much for your suggestion. We have checked.

l.116 - please correct the use of a capital letter.

Response: Thanks very much for your suggestion. We have checked.

l.133 - please correct the literature reference - name needed.

Response: Thanks very much for your suggestion. We have added name.

l.154-160 - models without any description (symbols?) are useless. Please comment more widely on the models or remove them from the text.

Response: Thanks very much for your suggestion. We have added symbols.

l.154-155 - models (1) and (2) are illegible; please use smaller letters in the indexes.

Response: Thanks very much for your suggestion. We have revised.

l.160 - please correct errors in formula (4).

Response: Thanks very much for your suggestion. We have corrected.

l.167-168 - please revise the content; ultrasound covers a much wider range of frequencies (above 100 kHz) and intensities (below 1 W/cm²).

Response: Thanks very much for your suggestion. We have revised.

l.185 - porous structures have high acoustic resistance; please verify the statement in the sentence.

Response: Thanks very much for your suggestion. We have verified the statement.

l.221 - please correct the unit notation.

Response: Thanks very much for your suggestion. We have corrected.

l.233 - first use of the abbreviation FD; please explain.

Response: Thanks very much for your suggestion. We have explained in Line 38.

l.295 - please verify and correct the information on drying time.

Response: Thanks very much for your suggestion. We have verified and corrected drying time.

Table 1 - please organize alphabetically by first column.

Response: Thanks very much for your suggestion. We have revised Table 1.

l.319 - please verify use of capital letter.

Response: Thanks very much for your suggestion. We have verified.

l.324 - first use of the abbreviation UHP; please clarify.

Response: Thanks very much for your suggestion. We have clarified in Line 319.

l.341 - unnecessary period.

Response: Thanks very much for your suggestion. We have deleted.

l.349 - parameter a* refers to color; please rephrase.

Response: Thanks very much for your suggestion. We have rephrased.

l.335 - revise and correct notation of arbitrary unit.

Response: Thanks very much for your suggestion. We have revised.

l.358 - necessary clarification of "W/L" designation (watt per liter?).

Response: Thanks very much for your suggestion. We have revised.

l.363 - unnecessary abbreviation in parentheses (Tg); it is used only once.

Response: Thanks very much for your suggestion. We have deleted.

Table 2 - please arrange alphabetically by first column.

Response: Thanks very much for your suggestion. We have arranged.

l.404 - the information on reducing time is meaningless, because the authors do not report the initial time. Reword the sentence.

Response: Thanks very much for your suggestion. We have revised.

409-410 - use italic in Latin name.

Response: Thanks very much for your suggestion. We have revised.

l.423 - revise and correct the use of a capital letter.

Response: Thanks very much for your suggestion. We have revised.

l.427 - the explanation of the FD25-FIRD2 designation is missing; please clarify.

Response: Thanks very much for your suggestion. We have clarified in Line 486.

Table 3 - please organize alphabetically by first column; insert 2 as superscript (Main results - Banana); correct use of capital letter in unit (kW).

Response: Thanks very much for your suggestion. We have revised.

l.467 and the next - please standardize the way of units of measurement are written; write with "/" or write with exponent of a power.

Response: Thanks very much for your suggestion. We have revised.

l.470 - please clarify the abbreviation ZP; it appears only in this line of text.

Response: Thanks very much for your suggestion. We have deleted.

Figure 7 - the caption on photos (b) and (d) is illegible; include a clear linear scale; please improve the quality or remove the illegible part of the photo.

Response: Thanks very much for your suggestion. We have revised.

l.483 - remove "di=" and "hi=".

Response: Thanks very much for your suggestion. We have deleted.

l.500 - first occurrence of the abbreviation HPD; please clarify.

Response: Thanks very much for your suggestion. We have clarified.

l.525 - please verify and correct the name of the reaction.

Response: Thanks very much for your suggestion. We have corrected.

Table 4 - please organize alphabetically by first column; please use capital letters correctly; please verify and standardize notation of units in column 2 (e.g. The initial moisture content - Banana).

Response: Thanks very much for your suggestion. We have revised.

Table 5 - please organize alphabetically by first column.

Response: Thanks very much for your suggestion. We have revised.

l.592 - the freeze drying technique has advanced significantly and the authors should verify/correct the theses posed in this sentence.

Response: Thanks very much for your suggestion. We have revised.

References - please verify the notation of journal names (abbreviated names / full names)

Response: Thanks very much for your suggestion. We have revised.

l.636 - please verify and correct the names of authors in the publication.

Response: Thanks very much for your suggestion. We have revised.

l.676 and the next - please verify the notation of the journal title (capital letters).

Response: Thanks very much for your suggestion. We have revised.

Has permission been obtained from the authors to use Figures 5,6,7,8 in this publication?

Response: Thanks very much for your suggestion. We have obtained and sent Editor by the email.

Reviewer 3 Report

Comments and Suggestions for Authors

This interesting topic is definitely worth a review. The main benefit of this work lies in the collection of references, most of which are clearly presented in tables. 

Nevertheless, there are still significant shortcomings in the paper that need to be rectified before publication: 

The introduction starts with very general claims, on the other hand the cited references are too specific.

l. 90: what should be meant by "resolution" here?

ll. 93..: not only freezing is decisive for ice crystal size but especially the storage conditions between freezing and drying.

ll.106-108: the temperature profiles are dependent on sample geomtry and the heat transfer mechansisms (contact/conduction/radiation)

ll. 107, 109, 119: the temperatures and states collapse temperatur, glass transistion temperature, eutectic point should be defined and described.

ll. 113-117: can't be understood, rephrase

eq. 1-4: equations need to be explained, also the symbols used!

ll. 168..: although possible  us applications are widespread in academic research and literature, the actual use in food production is limited to niche products

ll. 170..: not the waves are compressed/expanded but the medium is compressed/expanded by the waves!

ll. 185-190: here it stays absolutely unclear, if cavitation produces desired or undesired effects

figs. 3,4,6: these figures are comic like and do not show anything

ll. 200..: reflection, absorption and penetration are not special at all for IR

l.211: when IR is applied in freeze drying at vacuum conditions, air cannot carry the vapour!

ll. 238-244: meanwhile the main application of PF is the pretreatment of potatoes (and vegetables) in the chips/french fries production. This should be stated here.

chapter 3.1 to 3.4: the description of these important technologies is too superficial!

l. 272: the authors should explain atmospheric freeze drying, before. Why no vacuum is needed here?

ll. 282+301: the stated power per volume is relatively to what volume (sample/applicator?)

l. 308: stating the power without referencing the sample volume od mass is meaningless!

chapter 4.2.: why UHP is not included separately as pretreatment in this review?

tables 1- 4: the drying conditions can't be compared (different units....)

why there isn't any table for PEF?

Comments on the Quality of English Language

This interesting topic is definitely worth a review. The main benefit of this work lies in the collection of references, most of which are clearly presented in tables. 

Nevertheless, there are still significant shortcomings in the paper that need to be rectified before publication: 

The introduction starts with very general claims, on the other hand the cited references are too specific.

l. 90: what should be meant by "resolution" here?

ll. 93..: not only freezing is decisive for ice crystal size but especially the storage conditions between freezing and drying.

ll.106-108: the temperature profiles are dependent on sample geomtry and the heat transfer mechansisms (contact/conduction/radiation)

ll. 107, 109, 119: the temperatures and states collapse temperatur, glass transistion temperature, eutectic point should be defined and described.

ll. 113-117: can't be understood, rephrase

eq. 1-4: equations need to be explained, also the symbols used!

ll. 168..: although possible  us applications are widespread in academic research and literature, the actual use in food production is limited to niche products

ll. 170..: not the waves are compressed/expanded but the medium is compressed/expanded by the waves!

ll. 185-190: here it stays absolutely unclear, if cavitation produces desired or undesired effects

figs. 3,4,6: these figures are comic like and do not show anything

ll. 200..: reflection, absorption and penetration are not special at all for IR

l.211: when IR is applied in freeze drying at vacuum conditions, air cannot carry the vapour!

ll. 238-244: meanwhile the main application of PF is the pretreatment of potatoes (and vegetables) in the chips/french fries production. This should be stated here.

chapter 3.1 to 3.4: the description of these important technologies is too superficial!

l. 272: the authors should explain atmospheric freeze drying, before. Why no vacuum is needed here?

ll. 282+301: the stated power per volume is relatively to what volume (sample/applicator?)

l. 308: stating the power without referencing the sample volume od mass is meaningless!

chapter 4.2.: why UHP is not included separately as pretreatment in this review?

tables 1- 4: the drying conditions can't be compared (different units....)

why there isn't any table for PEF?

Author Response

Responses to Reviewer: 

The introduction starts with very general claims, on the other hand the cited references are too specific.

Response: Thanks very much for your suggestion. We have removed this paragraph from the introduction. Nowadays, consumers are increasingly demanding quality and nutrition from their food products, which requires researchers and the food industry to constantly develop new technologies to produce the best possible quality products with improved shelf life and to find product characteristics that are directly related to consumer satisfaction.

  1. 90: what should be meant by "resolution" here?

Response: Thanks very much for your suggestion. We have replaced "resolution" with "desorption".

  1. 93..: not only freezing is decisive for ice crystal size but especially the storage conditions between freezing and drying.

Response: Thanks very much for your suggestion. We have rewritten the passage.

ll.106-108: the temperature profiles are dependent on sample geomtry and the heat transfer mechansisms (contact/conduction/radiation)

Response: Thanks very much for your suggestion. We have rewritten the passage. It is worth noting that the temperature profiles are dependent on sample geomtry and the heat transfer mechansisms (contact/conduction/radiation). During the drying process, the temperature in different regions of the sample varies with the position of the sample and decreases from the surface to the sublimation interface [17].

  1. 107, 109, 119: the temperatures and states collapse temperature, glass transistion temperature, eutectic point should be defined and described.

Response: Thanks very much for your suggestion. We have defined and described in Line 111-121.

  1. 113-117: can't be understood, rephrase

Response: Thanks very much for your suggestion. We have rewritten the passage. Desorption drying is completed after sublimation drying, the stage is relatively slow, fruits and vegetables in the internal porous structure of the residual bonded water in the high temperature conditions desorption to form free water, free water at high temperatures in the form of water vapor diffusion to the outside of the fruit and vegetables [13].

  1. 1-4: equations need to be explained, also the symbols used!

Response: Thanks very much for your suggestion. We have explained and used the symbols in Line 154-174.

  1. 168: although possible us applications are widespread in academic research and literature, the actual use in food production is limited to niche products

Response: Thanks very much for your suggestion. We have rewritten the passage. The use of ultrasound in academic research has become progressively more widespread and successful, but practical applications in food production are limited to niche products and need to be further expanded and developed. The use of ultrasound in the dehydration of fruits and vegetables enhances heat transfer, increases drying rates and reduces energy consumption.

  1. 170: not the waves are compressed/expanded but the medium is compressed/expanded by the waves!

Response: Thanks very much for your suggestion. We have rewritten the passage. At this point the medium is compressed and expanded by the ultrasonic waves, just as a sponge is repeatedly pressed and released.

  1. 185-190: here it stays absolutely unclear, if cavitation produces desired or undesired effects

Response: Thanks very much for your suggestion. We have rewritten the passage. Cavitation effect can expand the internal pores of fruits and vegetables, improve the migration speed of water in the drying process, which can effectively strengthen the drying process and shorten the time of freeze-drying [30].

figs. 3,4,6: these figures are comic like and do not show anything

Response: Thanks very much for your suggestion. We removed Figures 4 and 6, and Figure 3 was modified.

Figure 3. Ultrasound(a), infrared radiation(b), pulsed electric field(c), microwave(d) and freeze-drying(e) [31] sketch.

  1. 200: reflection, absorption and penetration are not special at all for IR

Response: Thanks very much for your suggestion. We have deleted the section "Infrared radiation is characterized by absorption, reflection and penetration".

l.211: when IR is applied in freeze drying at vacuum conditions, air cannot carry the vapour!

Response: Thanks very much for your suggestion. We have rewritten the passage. Infrared radiation directly penetrates the interior of fruits and vegetables, generating molecular vibration, and heating the interior of fruits and vegetables, the internal temperature is higher than the surface temperature, so that the temperature gradient within the fruits and vegetables and its internal moisture migration of the humidity gradient in the direction of the same direction, to promote the migration of water, mi-grate to the surface of the water is evaporated, to accelerate the drying process of fruits and vegetables [34].

  1. 238-244: meanwhile the main application of PF is the pretreatment of potatoes (and vegetables) in the chips/french fries production. This should be stated here.

Response: Thanks very much for your suggestion. We have added this section. PEF is also used in the production of potato and vegetable fries, in the process of french fries PEF can reduce the consumption of water and energy, shorten the processing time, increase the internal porosity of fried potatoes and accelerate the evaporation of water vapor [47]. After pretreatment of potatoes, it will reduce the content of reducing sugar in the finished potato chips and reduce the occurrence of browning phenomenon [48]. In addition PEF has a better cutting effect on potatoes, making the cut surface of potatoes smoother [47, 49].

chapter 3.1 to 3.4: the description of these important technologies is too superficial!

Response: Thanks very much for your suggestion. We have revised and added to this section Line 176-328.

  1. 272: the authors should explain atmospheric freeze drying, before. Why no vacuum is needed here?

Response: Thanks very much for your suggestion. We have added this section.

Atmospheric freeze drying is the sublimation of ice at atmospheric pressure, where a water vapor pressure difference is created between the product and the air, which is the driving force that promotes ice sublimation [63]. Atmospheric freeze drying is an alternative technology to vacuum freeze drying due to its lower cost and continuous drying process [64]. Atmospheric pressure freeze drying has a long drying process due to the low vapor diffusion rate at atmospheric pressure and the blocked internal heat mass transfer, which limits its practical application [63]. Therefore, certain methods to enhance this process are necessary. The auxiliary treatment of ultra-sound can improve this disadvantage and the application of ultrasound in the atmospheric pressure freeze drying of fruits and vegetables has been explored by researchers in recent years and good results have been achieved in this area.

  1. 282+301: the stated power per volume is relatively to what volume (sample/applicator?)

Response: Thanks very much for your suggestion. We have added this section in Line 362 and 370.

  1. 308: stating the power without referencing the sample volume od mass is meaningless!

Response: Thanks very much for your suggestion. We have added this section in Line 493, 501, 502, 506, 510, 515, 521, 528 and 533.

chapter 4.2.: why UHP is not included separately as pretreatment in this review?

Response: Thanks very much for your suggestion. We have added this section (3.5. Ultra high pressure) in Line 318-328.

tables 1- 4: the drying conditions can't be compared (different units....)

Response: Thanks very much for your suggestion. We have revised the contents in Table.

why there isn't any table for PEF?

Response: Thanks very much for your suggestion. We have added this section as shown in Table 4.

Round 2

Reviewer 1 Report

Comments and Suggestions for Authors

The manuscript has been revised almost in accordance with the reviewer's comments. Request to add information on the cost of purchasing the necessary equipment to use assisted sublimation drying techniques. I meant to try to demonstrate whether the purchase of microwave, pulsed electric field, infrared, ultrasound and ultrasound assisted atmospheric pressure equipment alone would realistically reduce the cost of freeze-drying and whether it is at all possible under industrial conditions to apply the assistive techniques discussed in the manuscript. Please add this information, in short form, to your manuscript.

Author Response

Responses to Reviewer: 

The manuscript has been revised almost in accordance with the reviewer's comments. Request to add information on the cost of purchasing the necessary equipment to use assisted sublimation drying techniques. I meant to try to demonstrate whether the purchase of microwave, pulsed electric field, infrared, ultrasound and ultrasound assisted atmospheric pressure equipment alone would realistically reduce the cost of freeze-drying and whether it is at all possible under industrial conditions to apply the assistive techniques discussed in the manuscript. Please add this information, in short form, to your manuscript.

Response: Thanks very much for your suggestion. Most of the literature is oriented towards increasing the rate of drying, with little consideration of the economic cost of purchasing capital for the equipment. The most significant cost of a freeze dryer is the initial cost, which is the investment in raw materials and equipment, rather than the utilization of the freeze dryer, whereas the operating cost is relatively low. The application of physical fields in freeze drying reduces the drying time of food products, increases the drying rate, improves the productivity, and reduces the operating cost, which is higher per unit of production of a laboratory scale freeze dryer as compared to a freeze dryer in industry.

Reviewer 3 Report

Comments and Suggestions for Authors

All comments of the reviewer have been addressed by the authors and most of them adquately integrated in the manuscript.

Author Response

Responses to Reviewer: 

All comments of the reviewer have been addressed by the authors and most of them adquately integrated in the manuscript.

Response: Thanks very much for your comments and helping us to improve the manuscript.